# Disentangling the role of Africa in the global spread of H5 highly pathogenic avian influenza

Alice Fusaro [1]*, Bianca Zecchin [1], Bram Vrancken [2], Celia Abolnik[3], Rose Ademun[4], Abdou Alassane[5], Abdelsatar Arafa[6], Joseph Adongo Awuni[7], Emmanuel Couacy-Hymann[8], M.' Bétiégué Coulibaly[9], Nicolas Gaidet[10], Emilie Go-Maro[11], Tony Joannis[12], Simon Dickmu Jumbo[13], Germaine Minoungou[14], Clement Meseko [12], Maman Moutari Souley[5], Deo Birungi Ndumu [4], Ismaila Shittu[12], Augustin Twabela [15], Abel Wade[16], Lidewij Wiersma[17], Yao P. Akpeli[11], Gianpiero Zamperin[1], Adelaide Milani[1], Philippe Lemey[2] & Isabella Monne[1]*

The role of Africa in the dynamics of the global spread of a zoonotic and economically-important virus, such as the highly pathogenic avian influenza (HPAI) H5Nx of the Gs/GD lineage, remains unexplored. Here we characterise the spatiotemporal patterns of virus diffusion during three HPAI H5Nx intercontinental epidemic waves and demonstrate that Africa mainly acted as an ecological sink of the HPAI H5Nx viruses. A joint analysis of host dynamics and continuous spatial diffusion indicates that poultry trade as well as wild bird migrations have contributed to the virus spreading into Africa, with West Africa acting as a crucial hotspot for virus introduction and dissemination into the continent. We demonstrate varying paths of avian influenza incursions into Africa as well as virus spread within Africa over time, which reveal that virus expansion is a complex phenomenon, shaped by an intricate interplay between avian host ecology, virus characteristics and environmental variables.

[1] Department of Comparative Biomedical Sciences, Istituto Zooprofilattico Sperimentale delle Venezie, Legnaro, Italy. [2] KU Leuven, Department of Microbiology and Immunology, Rega Institute, Leuven, Belgium. [3] Department of Production Animal Studies, Faculty of Veterinary Science, University of Pretoria, Pretoria, South Africa. [4] National Animal Disease Diagnostics and Epidemiology Center (NADDEC), Entebbe, Uganda. [5] Laboratoire Central de l'Elevage (LABOCEL), Niamey, Niger. [6] National Laboratory for Veterinary Quality Control on Poultry Production (NLQP), Animal Health Research Institute, Giza, Egypt. [7] Accra Veterinary Laboratory, Accra, Ghana. [8] Laboratoire Central de Pathologie Animale, Bingerville, Côte d'Ivoire. [9] Laboratoire National D'Appui au Développement Agricole, Abidjan, Côte d'Ivoire. [10] CIRAD, UPR GREEN, Montpellier, France. [11] Laboratoire Central Vétérinaire de Lomé, Lomé, Togo. [12] National Veterinary Research Institute, Vom, Nigeria. [13] Laboratoire National Vétérinaire (LANAVET), Garoua, Cameroon. [14] Laboratoire National d'Elevage de Ouagadougou, Ouagadougou, Burkina Faso. [15] Veterinary Laboratory of Kinshasa, Kinshasa, Democratic Republic of the Congo. [16] Laboratoire National Vétérinaire (LANAVET), Yaoundé, Cameroon. [17] Laboratory Unit of the Emergency Prevention System (EMPRES), Food and Agriculture Organization of the United Nations (UN-FAO), Rome, Italy. *email: afusaro@izsvenezie.it; imonne@izsvenezie.it

The highly pathogenic avian influenza (HPAI) virus of the H5N1 subtype was first identified in 1996 in the Chinese province of Guangdong and since then it has spread to other continents on multiple occasions. The emergence and global dissemination of this HPAI virus (hereafter named the Gs/GD lineage) has resulted in damages of unprecedented proportions to the poultry industry, impacting on the subsistence of the affected rural populations, national economies and international trade of live poultry and poultry products[1,2]. While unexpected for an avian influenza virus (AIV), the Gs/GD lineage also proved to have a substantial impact on human health[3], as highlighted by the 860 human infections including 454 deaths that have been reported as of April 9, 2019[4].

The sustained global circulation of this lineage has led to the diversification of the *hemagglutinin* (*HA*) gene segment into ten distinct clades (0–9), which subsequently evolved into second, third, fourth and fifth order subclades[5]. In the last 2 decades, three of the four trans-continental epidemic waves of the Gs/GD lineage also spread to Africa. Specifically, the African avian population has been infected by strains from clades 2.2 (H5N1 subtype), 2.3.2.1c (H5N1 subtype) and 2.3.4.4—group B (H5N8 subtype). The Gs/GD lineage, clade 2.2, was introduced for the first time in Africa in late 2005, affecting domestic birds in West Africa and Egypt[6]. In Egypt, the virus became endemic and since then has further evolved into clades 2.2.1, 2.2.1.1a and 2.2.1.2[5]. In January 2015, seven years after the eradication of the HPAI H5N1, a new clade, 2.3.2.1c, was introduced into the West African poultry population, where it is still occasionally causing outbreaks[7]. The last incursion of the Gs/GD lineage, clade 2.3.4.4 —group B (2.3.4.4-B), into Africa occurred in November 2016, and for the first time the epidemic spread to several countries in northern, western, eastern, central and southern Africa. Egypt, Tunisia and Nigeria were the first countries reporting the disease[8–10], followed by Niger, Cameroon and Uganda[11–13]. In spring 2017, the virus reached the Democratic Republic of the Congo (DR Congo)[14], Zimbabwe and South Africa[11,15] and in February 2019 new cases were reported in Namibia[16]. Currently, Gs/GD HPAI H5Nx poses a substantial threat to the poultry population in several African countries, and distinct clades are co-circulating in West Africa (clade 2.3.2.1c and 2.3.4.4-B) and Egypt (clades 2.2.1.2 and 2.3.4.4-B)[17,18] (Supplementary Fig. 1).

Despite the sustained circulation of Gs/GD HPAI H5Nx in Africa, the relevance of this continent in the dynamics of the global spread of this zoonotic and economically important virus is unknown. In this study, we analyse more than 1200 sequences, of which 40 are newly generated, to compare the phylogeographic patterns of the viruses collected during the three epidemic waves, on both a global and a continental (Africa) scale. We characterise the spatiotemporal patterns of virus diffusion to/from and within Africa and investigate the role that poultry trade and wild bird migration may have played in the spread of the virus. This contributes to increase our predictive capability of virus gene flows, which can be instrumental for epidemic preparedness. We reveal that Africa acted mainly as an ecological sink of the Gs/GD HPAI H5Nx viruses and show varying paths of AIV introduction into the continent over time. Importantly, we identify the African regions at high risk of incursion and of co-circulation of multiple clades, which can favour the emergence of viruses with pandemic potential, thus providing a baseline for improving future surveillance programmes.

## Results

### Datasets and missing data.
We analysed two datasets—a global and an African dataset—for each HPAI H5Nx clade of the Gs/GD lineage that reached the African continent: clade 2.2 (2005–2011), clade 2.3.2.1c (2011–2017) and clade 2.3.4.4-B (2014–2018). For the African datasets, we used all the available sequences of the viruses collected on the African continent, while for the global datasets three different subsampling strategies (see Supplementary Methods) were used to mitigate and assess the impact of potential sampling biases.

For the global analyses, we defined nine discrete regions—West Europe, East Europe, The Middle East, East Asia, North-Central Asia, South Asia, West Africa, East-Central Africa and South Africa—and four host types—domestic Galliformes, domestic Anseriformes, wild Anseriformes and other wild bird species. This subdivision enabled us to have well represented categories for each geographical region and host type trait. For the analyses of the African datasets, the discrete regions correspond to the country of collection, while host types were not incorporated because the majority of available sequences are from domestic birds.

It is important to consider that disease surveillance, outbreak reporting and sequencing efforts vary considerably between countries. The number of reported HPAI H5Nx outbreaks in domestic birds corresponds well with the intensity and distribution of poultry production (Supplementary Fig. 2). On the other hand, passive and active surveillance in wild bird populations appears to be very limited, except for North-Central Asia, Europe and South Africa, for which 66% (159/249), 49% (1569/3173) and 42% (78/185) of the reported HPAI H5Nx outbreaks are from wild birds. As a result, with the exception of these three geographic areas, there are many more reported outbreaks from domestic (98%) than wild (2%) birds[19]. In West Africa, despite the occurrence of several HPAI H5Nx introductions and the presence of large congregation sites of wild waterbirds, to date few outbreaks (1%) have been reported in wild species[19].

The *HA* genes for 29% of the viruses from the reported outbreaks in the geographic areas under study were available and the number of sequences was generally proportional to the number of reported outbreaks in each discrete geographic area of this study, although not constant across time, but varying from 3% in 2018 to 59% in 2008 (Supplementary Fig. 2). For the African continent, the *HA* gene of 35% of the viruses from the reported outbreaks have been sequenced and the proportion of *HA* sequences per reported outbreaks varies from 9% in 2018 to 87% in 2007 (Supplementary Fig. 2).

### The global sources of the African HPAI H5Nx viruses.
To shed light on the potential origins of HPAI H5Nx viruses that reached the African continent, we performed discrete phylogeographic analyses of the *HA* gene of the HPAI H5Nx epidemic waves caused by clades 2.2, 2.3.2.1c and 2.3.4.4-B and reconstructed their global dissemination. For each clade, we compared the results obtained from three differently down-sampled datasets (epi-based, tree-based and random, see Supplementary Methods). Since the overall migration pattern is consistent across the down-sampling strategies, we described here only the results obtained for the selection of representative sequences by different epidemiological characteristics (epi-based datasets), which has the most balanced distribution of samples among locations and hosts. 75% (clade 2.3.4.4-B) to 100% (clades 2.2 and 2.3.2.1c) of the epi-based transition events (BF > 5) were identified in at least one other down-sampled dataset (Supplementary Table 1).

According to the most probable location at the root of the maximum clade credibility (MCC) trees, all clades emerged in East or North-Central Asia. Specifically, we found a maximum root state probability for North-Central Asia and East Asia for clade 2.2 and 2.3.4.4-B, respectively, and they were also the only locations with non-zero posterior probability as the origin of

2.3.2.1c clade (Supplementary Figs. 3–5). From these geographic areas, the three lineages subsequently spread southward to South Asia and westwards to the Middle East, Europe and Africa (Fig. 1). However, the routes and number of virus introductions into the African continent vary across epidemic waves: while Europe seems to have been the key geographical source for clade 2.2 viruses found in Africa, clade 2.3.2.1c appears to have been introduced into Africa from the Middle East and South Asia, while clade 2.3.4.4-B from North-Central Asia (Fig. 1).

Specifically, during the first wave (2005/2006) we identified four distinct H5N1 lineages within clade 2.2 in West Africa and one in Egypt, suggesting the occurrence of at least five separate introductions into the continent, four during the first half of 2006 and one in 2008 (Supplementary Fig. 3). Given that only East Europe has a BF support >5 as the origin of clade 2.2 viruses in Africa (Fig. 1), and that the posterior origin location probability is >0.85 for East Europe for each of the identified introductions (Supplementary Fig. 3), East Europe represents the most likely origin of clade 2.2 viruses in Africa. From West Africa, the virus subsequently spread to East-Central Africa (BF = 493.89; PP = 0.99) and to the Middle East (BF = 6.89; PP = 0.52). These results were confirmed by the analyses of the three subsampled datasets (Supplementary Table 1). A more detailed phylogeographic reconstruction in continuous space corroborates this sequence of events and suggests that most of the virus incursions into Africa occurred from the area surrounding the Black Sea (Fig. 2). The time to the most recent common ancestor (tMRCA) estimates indicate that four of the identified introductions might have occurred in 2005/2006 (March 2005–April 2006) and one between July 2007 and March 2008. The long time period covered by the branches that separate the African lineages from their most closely related European strains prevents a more precise timing of the introductions (Fig. 3, Supplementary Table 2). The last outbreak caused by clade 2.2 was reported in West Africa in 2008, while in Egypt the virus is still entrenched in the poultry population.

The topology of the MCC tree of clade 2.3.2.1c (Supplementary Fig. 4) suggests two almost simultaneous introductions into West Africa from South Asia and the Middle East during the second H5N1 intercontinental spread (2014–2015). However, the relatively moderate Bayes factor support for either location as a source of HPAI in West Africa (Supplementary Table 1), added to the marked relatedness between the identified viruses, could mean that a single virus introduction from an unsampled location, followed by a diverging evolutionary event in Africa, cannot be ruled out (Fig. 1, Supplementary Fig. 4). The continuous phylogeographic analysis also shows two virus introductions in West Africa, but the limited availability of viral gene sequences, in particular from the Middle East and the area surrounding the Caspian and Black Seas, hampers our accurate reconstruction of the history of the spread (Fig. 2). Our estimation of the tMRCA indicates that the virus might have been introduced in Africa between May and November 2014 (Fig. 3). Unlike the first epidemic wave, this clade has been identified only in the western part of the African continent, where it is still reported by several countries.

The last epidemic wave was caused by the H5N8 subtype belonging to clade 2.3.4.4-B. We identified two separate virus incursions into West Africa and three into Egypt during winter 2016–2017 (Supplementary Fig. 5). Despite the extensive circulation of this strain in Europe that was also observed during the first epidemic wave, East Europe appears to have been the origin of the virus only for one of the introductions in Egypt (posterior probability = 0.95, Supplementary Fig. 4). The other virus incursions into Africa likely originate from North-Central Asia (posterior probabilities range from 0.37 to 0.96, Supplementary Fig. 5).

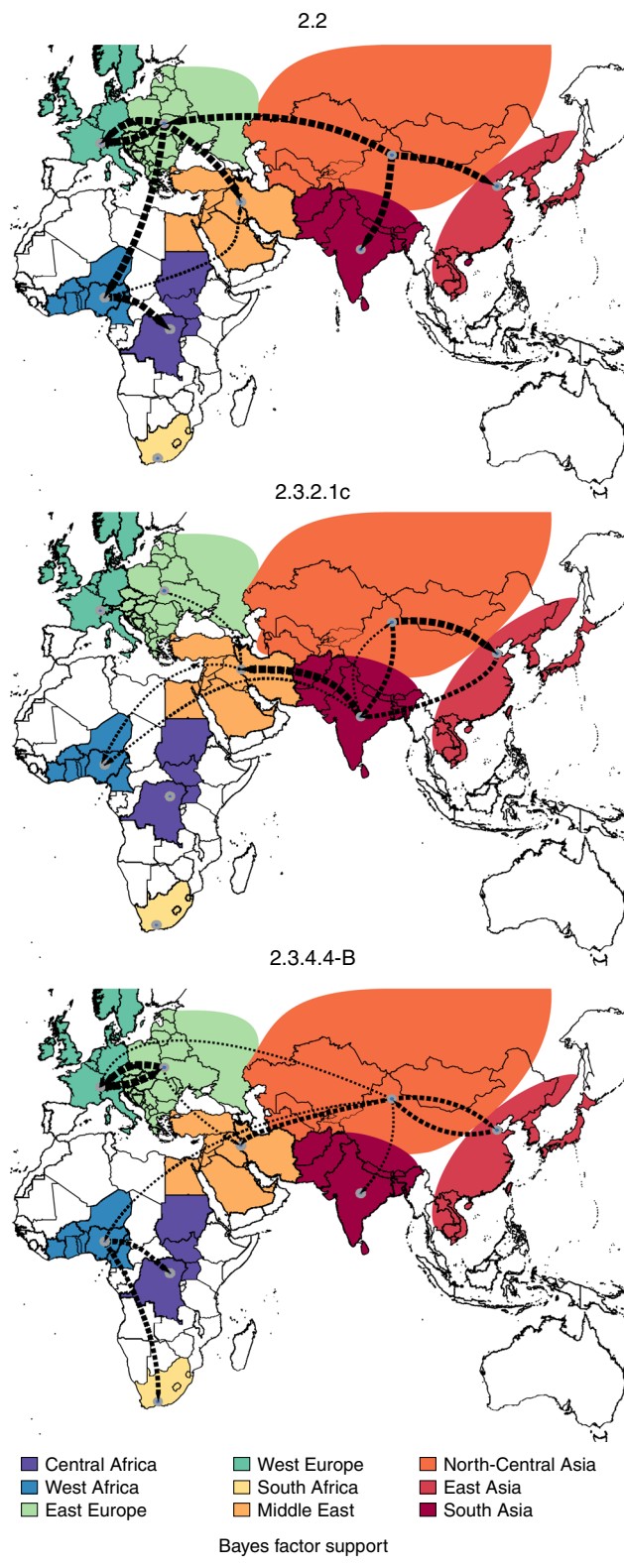

**Fig. 1** Global migration rates between geographic regions of the three HPAI H5Nx clades. Maps showing statistically supported non-zero rates (BF > 5) for clades 2.2, 2.3.2.1c and 2.3.4.4-B. Areas for each region type are labelled using the same colour in the annotated phylogenetic trees in Supplementary Figs. 3–5. The thickness of the dashed lines representing the rates is proportional to the relative strength by which rates are supported for the epi-based datasets shown in the Supplementary Table 1: very strong (BF > 150, thick lines), strong (20 < BF < 150, medium lines) and positive (5 < BF < 20, thin lines).

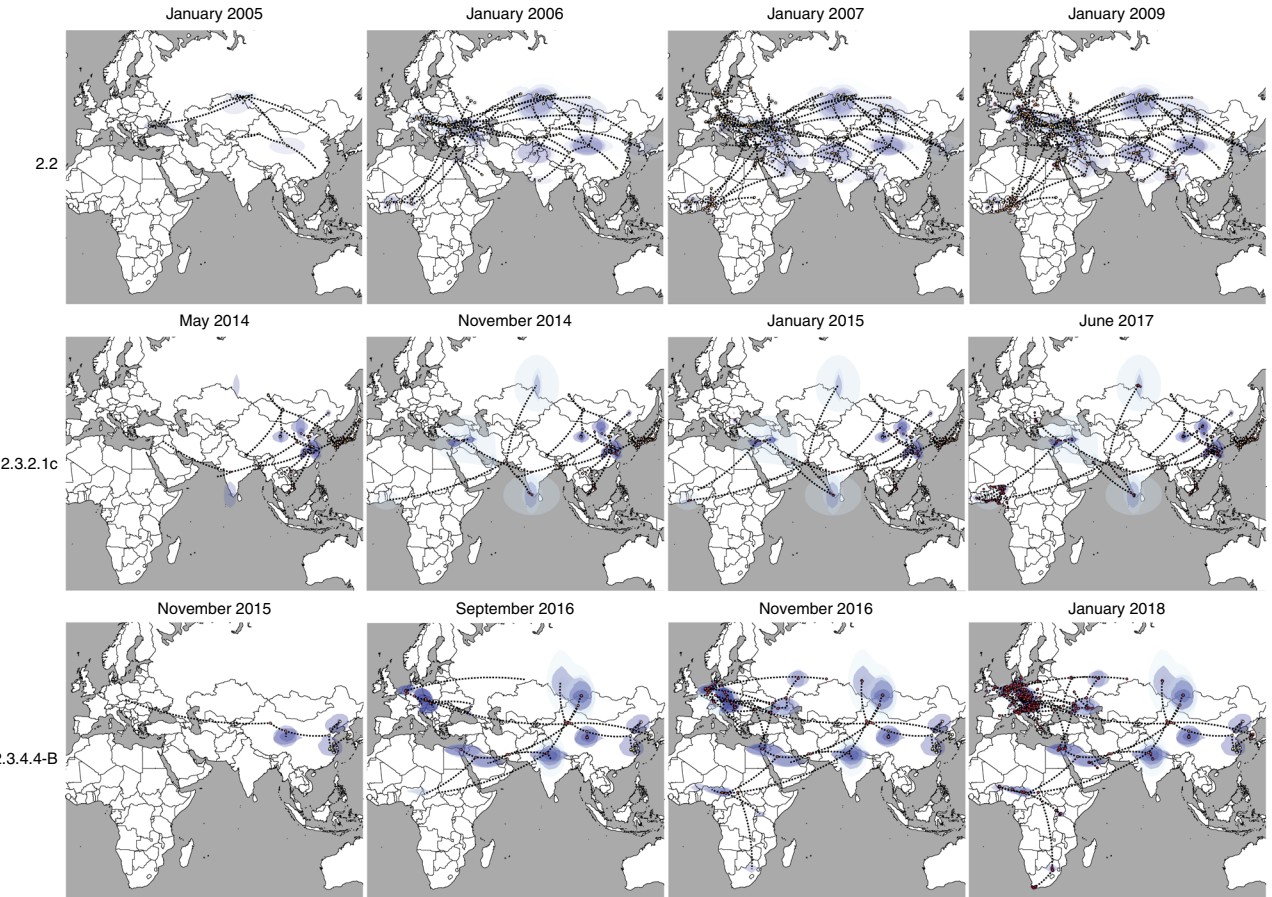

**Fig. 2** Global spatiotemporal dispersal of the three HPAI H5Nx clades. Dispersal patterns inferred using continuous phylogeographic analysis of the epi-based datasets are shown for four time slices for each of the three HPAI H5Nx clades. The black dashed lines and the dots represent part of the branches and the nodes of the MCC tree up to each of the indicated time. Dots are coloured according to the time (from yellow for the oldest to red for the youngest). Contours represent statistical uncertainty of the estimated locations at the internal nodes (95% credible contours based on kernel density estimates).

However, the uncertain origin and the long branches that separate the North-Central Asian viruses from their progeny in Africa are again suggestive of important data gaps (Fig. 1 and Supplementary Fig. 5). From West Africa, the virus spread to East-Central (posterior probability = 0.98) and South (posterior probability = 1) Africa (Supplementary Fig. 5). Phylogeographic reconstruction in continuous space suggests a westward virus spread from China to Europe, the Middle East and Africa (Fig. 2). We can only speculate that the virus spread from China to Mongolia, Siberia and west Russia, where the first European viruses were identified at the beginning of October 2016, and southwards to the Middle East and Africa. The data are not sufficiently informative to determine whether the identification of H5N8 viruses in different areas of the African continent (west, east-central and south) was a consequence of internal virus movement or whether they originated from separate virus introductions of similar variants. Also, the 95% HPD intervals of tMRCAs are wide: the virus might have been introduced in West Africa from April to September 2016 (95% HPD, January–November 2016) and from May to November 2016 (95% HPD, January–December 2016). Introductions into Egypt, on the other hand, likely occurred between June and December 2016 (95% HPD, March–December 2016) (Fig. 3, Supplementary Table 2).

**Virus spread within the African continent**. To explore how the virus spread within the African continent, we performed both discrete and continuous analyses. For all clades, West Africa was

the most important origin of the virus for the central, eastern and southern African countries. In particular, within West Africa, Nigeria was the most important point of virus introduction and was a central hub in the virus spread to other countries during the first and the second epidemic waves (Figs. 4 and 5 and Supplementary Figs. 6 and 7). Specifically, during the first wave we identified four virus introductions into Nigeria likely from East Europe, three during the winter of 2005–2006, which were followed by a rapid movement of the virus from the north-central to the southern area of the country and vice versa and a fourth incursion in 2008 (Figs. 4 and 5 and Supplementary Fig. 6), consistent with previous studies[20–24]. Between 2006 and 2007, six spillover events from Nigeria to Burkina Faso, Benin, Niger (two introductions), Togo, Sudan and further virus spread from Burkina Faso to Ivory Coast and from there to Ghana were identified (Supplementary Fig. 6). In addition, our discrete analysis indicates that, in most cases, viruses sampled from individual countries tended to cluster together, which is highly suggestive of considerable geographic structure among African clade 2.2 viruses.

During the second epidemic wave, Nigeria emerged again as the most important point of virus introduction in West Africa and the most important source for the other West African countries, like Burkina Faso (two introductions), Niger (four introductions) and Ivory Coast (one introduction). Burkina Faso was central to the virus diffusion into Ivory Coast and Ghana, while HPAI H5N1 entered Cameroon and Togo likely via Niger and Ghana, respectively (Figs. 4 and 5, Supplementary Fig. 7).

The pattern of virus diffusion within the continent during the last epidemic (clade 2.3.4.4-B) differs from that observed during the previous waves. For the first time, the Gs/GD lineage reached eastern and southern Africa where a high number of wild birds were affected. In spite of the sparse sampling, West Africa

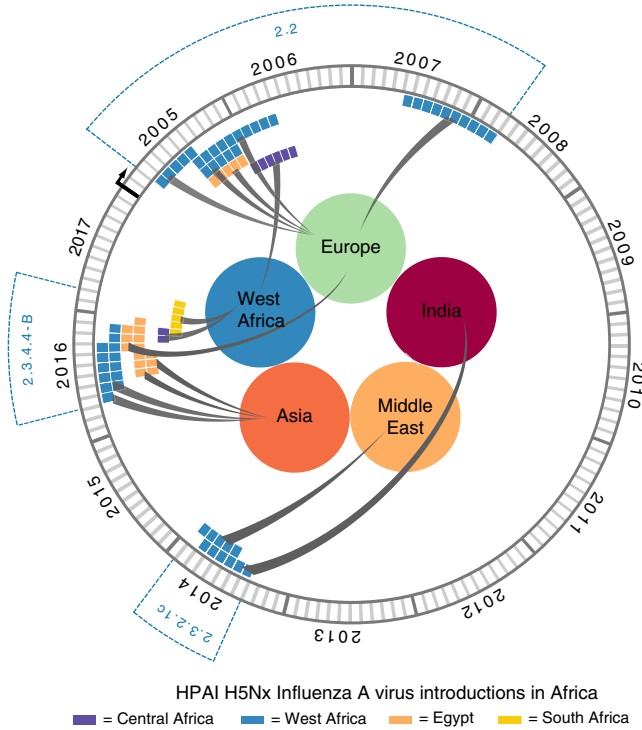

**Fig. 3** tMRCA estimated for each virus introduction in Africa. Coloured bars represent the mean tMRCAs (Supplementary Table 2, Global dataset) of each virus introduction in West Africa (blue), Egypt (orange), East-Central Africa (violet) and South Africa (yellow). Circles represent the area of origin of each virus introduction, based on the estimates summarised in the maximum clade credibility trees of each HPAI H5 clade.

(Cameroon, Niger and Nigeria) again acted as a central hotspot for the virus introduction and dissemination in the continent. This region experienced two virus incursions, likely in the second half of 2016, in Cameroon and Niger, where two co-circulating genetic groups were detected (Supplementary Fig. 8). As only a single sequence was available from Nigeria, its role during this epidemic wave cannot be assessed. Surprisingly, just one of the two groups detected in West Africa was identified in East-Central Africa (Uganda and subsequently DR Congo) and South Africa. Specifically, viruses from East-Central Africa were most closely related to the first group of viruses detected in Cameroon, Niger and Nigeria (WA-Introduction 1), while South African viruses clustered with the second group of West African samples identified in Cameroon and Niger (WA-Introduction 2) (Supplementary Fig. 8). As the Ugandan outbreaks occurred almost simultaneously with the Western African outbreaks, it is difficult to establish the direction of virus spread (from east to west or from west to east Africa) (Figs. 4 and 5) or to exclude the possibility of separate introductions from the same location. Sequencing of a wider number of samples could reveal the co-circulation of other variants in these areas of the continent and could uncover different transmission dynamics.

**Role of domestic and wild birds in virus spread.** To disentangle the role of poultry trade and wild bird migration in the spatial expansion of the three HPAI H5Nx clades, we performed a joint analysis of discrete host (domestic Galliformes, domestic Anseriformes, wild Anseriformes and other wild bird species) and continuous spatial diffusion for each of the three global epi-based datasets. For clade 2.2 and 2.3.4.4-B, wild Anseriformes showed the highest rate of spread; however, for clade 2.3.4.4-B the estimated transmission rate by wild Anseriformes substantially overlaps with that by domestic Galliformes. A similar pattern emerged for clade 2.3.2.1c, with the only difference being other wild species and domestic Galliformes the hosts with the highest rate (Fig. 6a). Interestingly, spread by domestic Anseriformes turned out to be the slowest in all epidemic waves. In particular, for clades 2.2 and 2.3.4.4-B a significantly higher rate of spread in

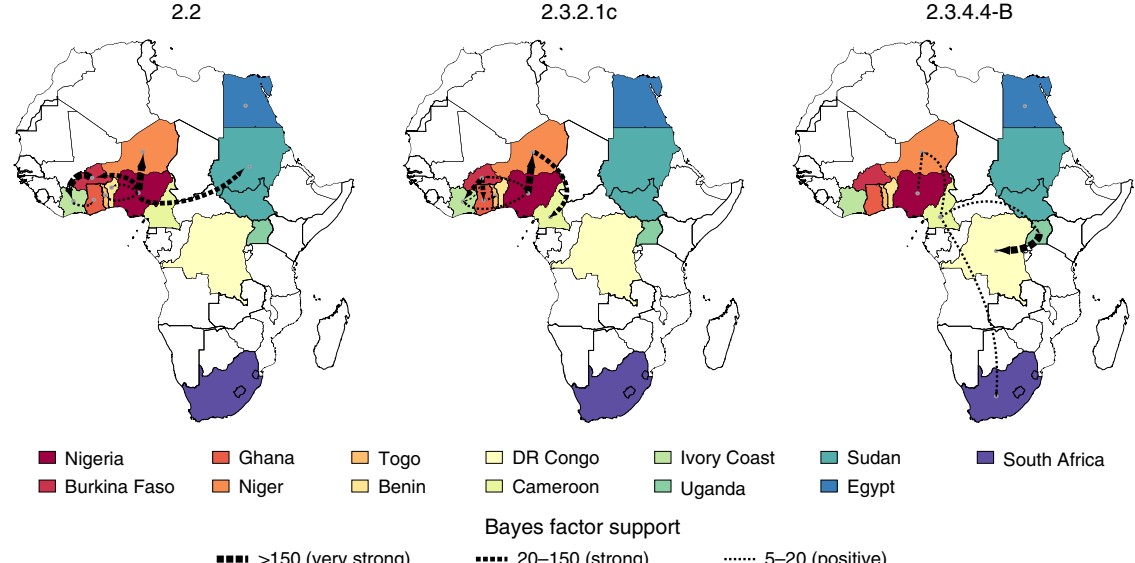

**Fig. 4** Migration rates between African countries of the three HPAI H5Nx clades. Maps showing statistically supported non-zero rates (BF > 5) for clades 2.2, 2.3.2.1c and 2.3.4.4-B. Each country is labelled by the same colour used in the annotated phylogenetic trees in Supplementary Figs. 6–8. The thickness of the dashed lines representing the rates is proportional to the relative strength by which rates are supported for the epi-based datasets shown in the Supplementary Table 1: very strong (BF > 150, thick lines), strong (20 < BF < 150, medium lines), and positive (5 < BF < 20, thin lines).

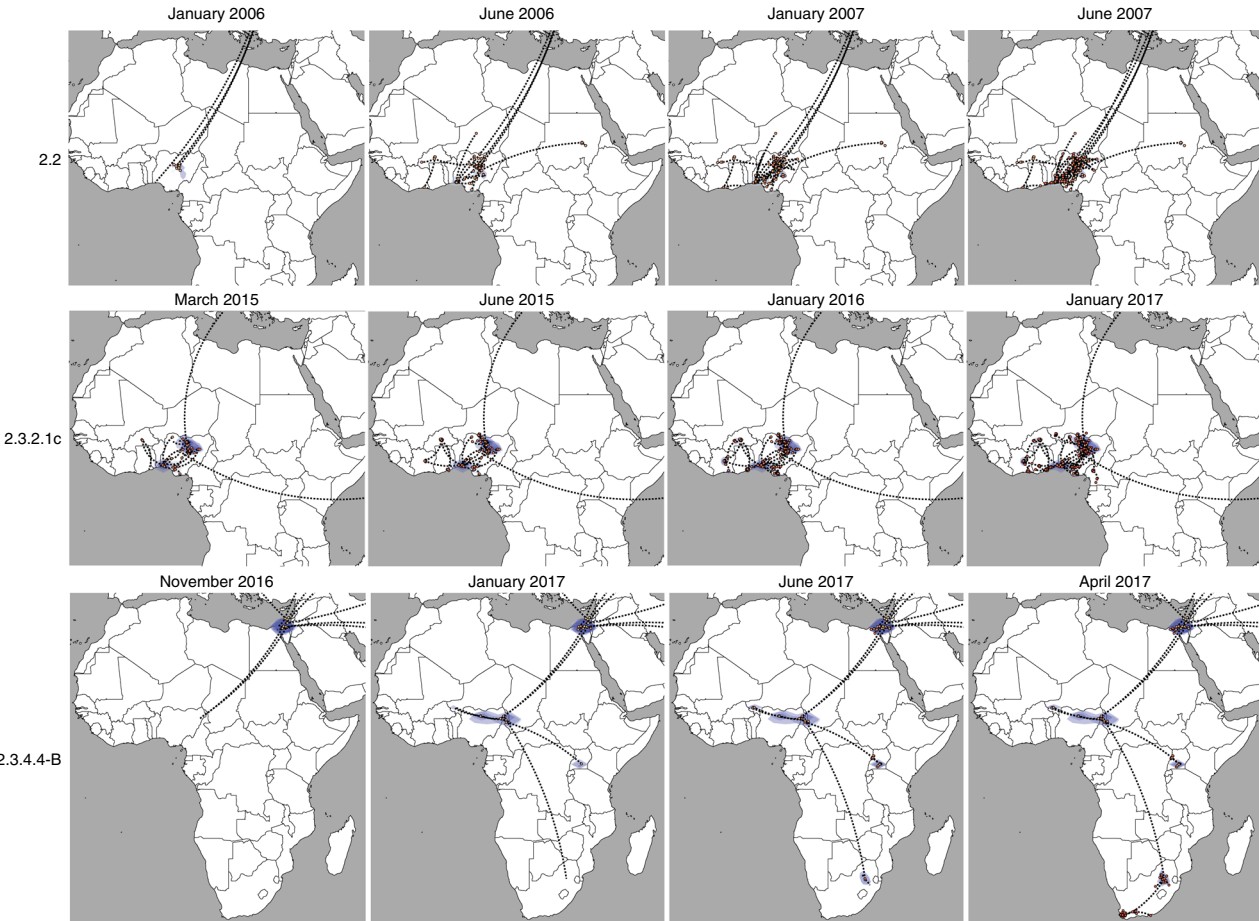

**Fig. 5** Spatiotemporal dispersal of the three HPAI H5Nx clades within the African continent. Dispersal patterns of H5Nx viruses in Africa, inferred using continuous phylogeographic analysis, are shown for four time slices for each of the three HPAI H5Nx clades. The black dashed lines and the dots represent part of the branches and the nodes of the MCC tree up to each of the indicated time. Dots are coloured according to the time (from yellow for the oldest to red for the youngest). Contours represent statistical uncertainty of the estimated locations at the internal nodes (95% credible contours based on kernel density estimates).

wild compared to domestic Anseriformes can be observed, indicating that the host contribution to virus diffusion is mainly linked to the degree of domestication rather than to the host order (Fig. 6a).

Although the three epi-based datasets were built to be fairly balanced in terms of sampling location, collection date and host (Supplementary Methods), the number of sequences from wild birds turned out to be very heterogeneous per region. To overcame this host skewed data for certain geographic areas, such as West and East-Central Africa, South Asia and the Middle East, for which the available sequences from wild birds ranged from 5% (West Africa) to 24% (the Middle East), we repeated the analyses by allowing only host species transitions from wild to domestic birds, as to consider the abundant evidence that during and after 2005, Gs/GD lineage introduction in poultry in multiple regions was associated with wild bird migration[12,13,15,25–36]. Using such enforcement, our estimates reveal a significantly higher rate of viral spread in wild birds compared to domestic ones (Fig. 6b). Specifically, during both first and third epidemic waves, wild Anseriformes contributed most to the virus expansion, while other wild bird species dominated in the diffusion during the second epidemic wave (Fig. 6b). Because the estimates for the other wild bird species are highly uncertain, we caution against drawing strong conclusions for the contribution of this host category to HPAI H5Nx spread.

We also explored the role of different host categories in virus introduction into Africa (Fig. 6c). Host constraint was set to prevent bias due to the heavy unbalanced data from wild and domestic birds for this continent. During the first epidemic wave, both wild and domestic birds seem to have contributed to virus introduction into the continent, while domestic Galliformes and wild Anseriformes appear to be mainly responsible for virus incursion into Africa during the second and third epidemic waves, respectively. However, we cannot exclude that this analysis could be affected by the lack of African viruses from wild birds, in particular for the first two epidemic waves. This biased sampling prevented us from exploring the host contribution to the virus diffusion within the African continent. However, given the wide and persistent circulation in poultry of clade 2.2 in West Africa (2006–2008) and Egypt (2006–present) and of clade 2.3.2.1c in West Africa (2015–present), poultry trade has likely been the major driver of virus spread of these two clades within Africa. For clade 2.3.4.4-B, however, our data indicate a potential contribution of wild birds in the virus spread within Africa. Several wild bird species were affected during this last wave, including African partial-migrants, like spur-winged goose and sacred ibis[11,37,38]. However, waterbird movements within Africa are poorly understood and are highly variable among species[39–42], making it difficult to assess their potential role in virus diffusion. Nor can we exclude that the viruses identified in west, east and south

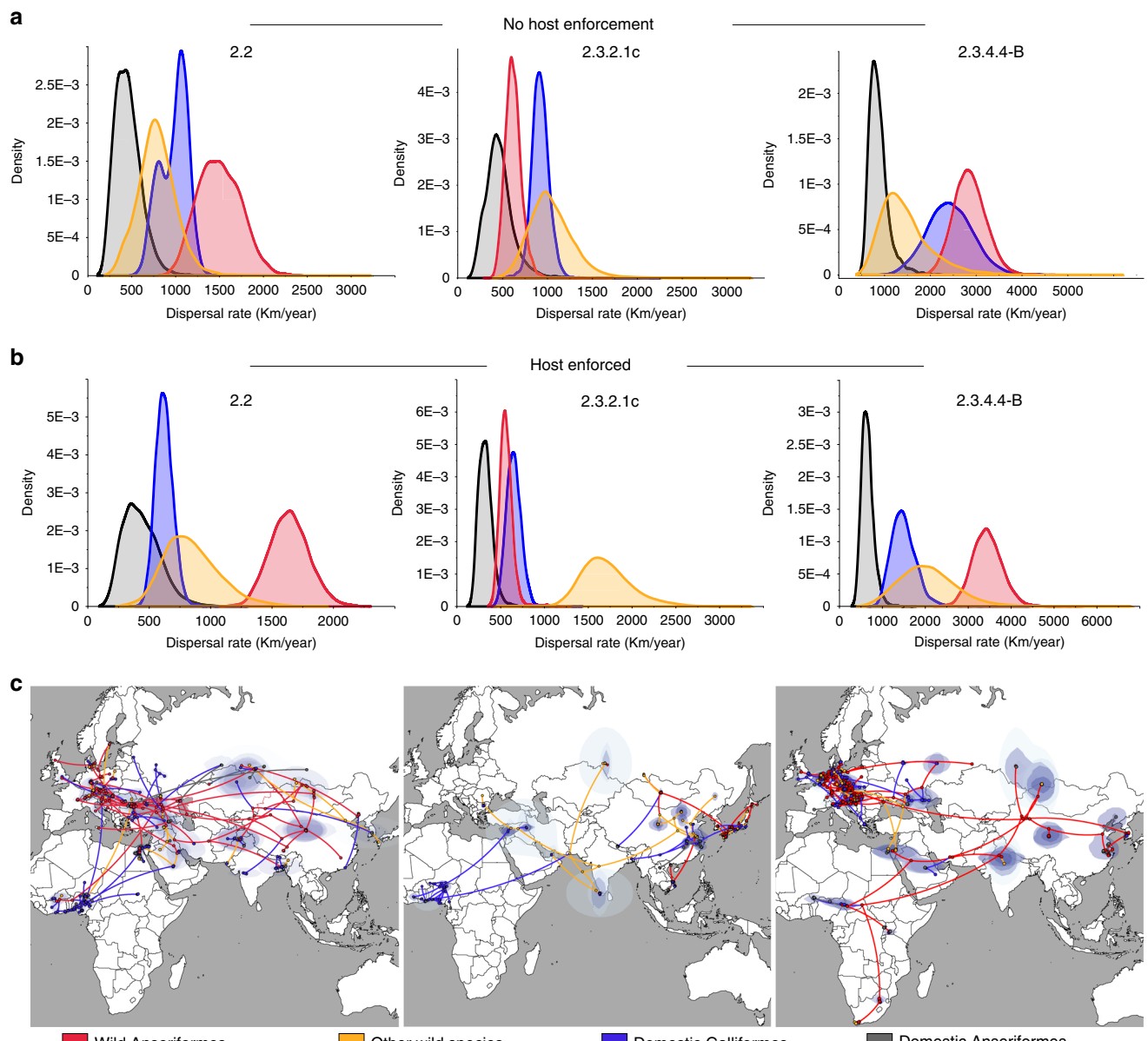

**Fig. 6** Contribution of different host types to HPAI H5Nx dissemination. Posterior dispersal rate distributions for each host (red—wild Anseriformes; yellow —other wild species; blue—domestic Galliformes; grey—domestic Anseriformes) obtained by the joint host analyses of the epi-based datasets of the three clades without any host enforcement (**a**) or imposing host species transitions from wild to domestic birds (**b**). **c** Dispersal patterns obtained from the epi-based datasets are shown for each of the three HPAI H5Nx clades. The lines and dots represent the branches and nodes of the MCC trees and are marked according to the most probable ancestral host trait as described above. Contours represent statistical uncertainty of the estimated locations at the internal nodes (95% credible contours based on kernel density estimates).

Africa derive from separate introductions of genetically similar viruses.

Previous studies demonstrated that extremely cold winters can influence wild bird migrations and modulate the wintering distribution of wild birds in the temperate regions[43,44]. Figure 7 shows the world temperature anomaly maps for the months of October, November and December of the years during which an intercontinental Gs/GD HPAI H5Nx spread was reported: 2005, 2009, 2014 and 2016[45]. The maps were obtained from the National Oceanic and Atmospheric Administration (NOAA)[46] and were created comparing the land and ocean surface temperatures of a given month to the average values for that month for the period 1901–2000. A positive anomaly (red) indicates that the observed temperature was warmer than the reference value, while a negative anomaly (blue) indicates that the

observed temperature was cooler than the reference value (Fig. 7). In 2005, a cold winter affected Europe for two consecutive months and this might have favoured the southern spread of the virus, as previously suggested by Ottaviani et al. [43]. Similarly, in October–December 2016 North-Central Asia, East Europe and the areas surrounding the Black and Caspian seas experienced a persistent and severe negative anomaly. On the contrary, in the other years similar anomalies were observed for a limited period of time or in a less extensive area.

However, based on the present knowledge of the ecology of wild migratory birds, temperature anomalies can influence the bird migration in the temperate and boreal regions. Differently, in Sub-Saharan Africa the main trigger for bird movements is the availability of food and water, which is affected by rainfall[39]. Thus, these temperature variations may explain the south-

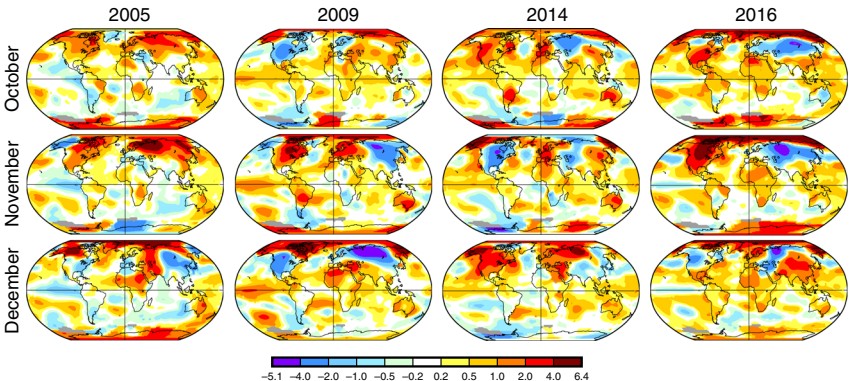

**Fig. 7** Word temperature anomaly maps. The temperature anomaly maps for the months of October, November and December of the years during which an intercontinental HPAI H5 spread was reported—2005, 2009, 2014 and 2016—were obtained from the National Oceanic and Atmospheric Administration (NOAA)[46] at https://www.ncdc.noaa.gov/sotc/global/201709 by comparing the land and ocean surface temperatures of a given month to the average values for that month for the period 1901–2000. A positive anomaly (red) indicates that the observed temperature was warmer than the reference value, while a negative anomaly (blue) indicates that the observed temperature was cooler than the reference value. Temperature anomaly in degrees Celsius.

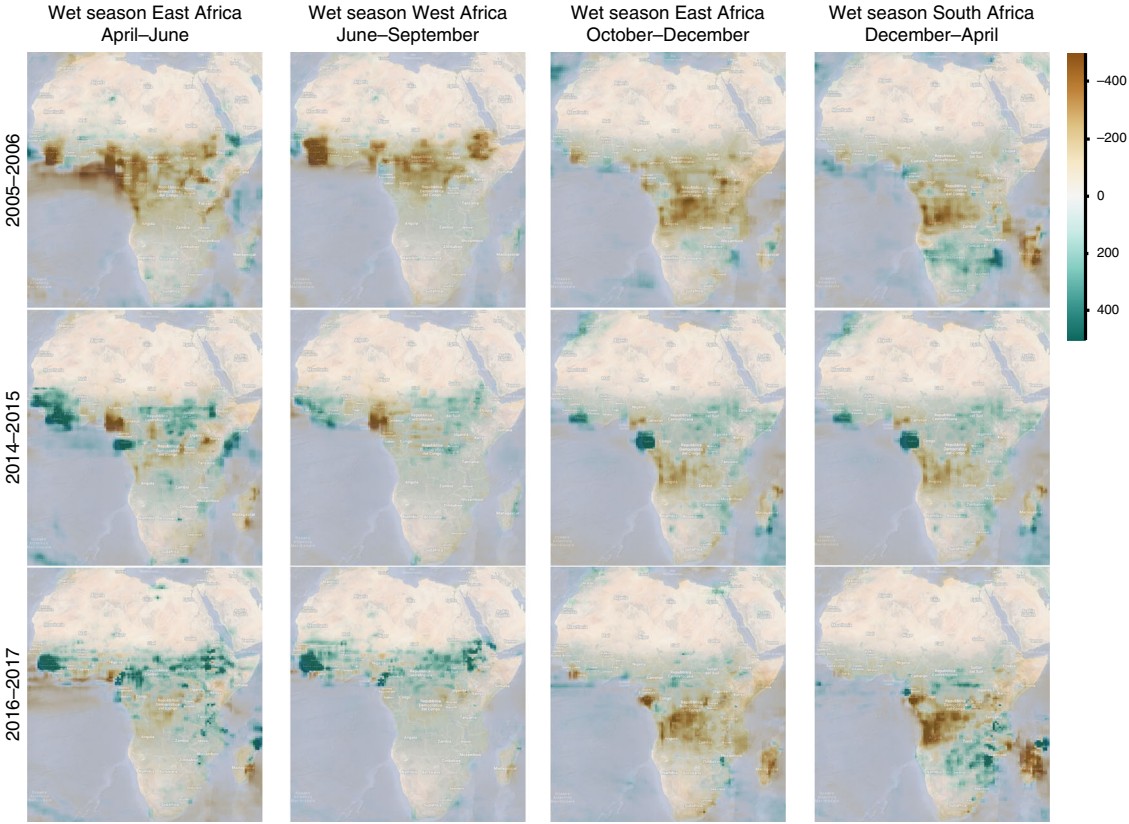

**Fig. 8** Precipitation anomaly maps for the African continent. Total precipitation rates difference from the 1981 to 2018 baseline mean for the wet seasons in east Africa (April–June and October–December), West Africa (June–September) and South Africa (December–April) for years 2005–2006, 2014–2015 and 2016–2017. Scale bar shows the precipitation (in mm) difference from average. Maps were obtained from Climate Engine[49] available at https://app.climateengine.org. Data Source: MERRA2 ~50-km (0.5° × 0.625°) daily dataset (NASA).

western migration of wild birds, and the associated virus spread, from North-Central Asia to the Middle East and northern Africa, including Egypt, but seem unlikely to be the causative factor for wild bird movement into West Africa. In Africa, when rain falls in arid and semi-arid areas, temporary productive wetlands may appear and attract large numbers of waterbirds[47]. We reviewed the total precipitation rate difference from the 1981 to 2018 mean for the wet seasons in east Africa (April–June and

October–December), West Africa (June–September) and South Africa (December–April) for each of the years reporting Gs/GD HPAI H5Nx virus incursions (Fig. 8, data from Climate Engine, 2018[48,49]). Noteworthy is the fact that in 2016, when wild birds were likely involved in the virus introduction and further spread in the African continent, Sub-Saharan regions experienced a remarkably wetter rainy season than usual, which might have created a favourable ecological condition for virus introduction

and spread[39]. The following anomalous drought, which affected central Africa during the October–December 2016 wet season, coupled with the abundant rainfall in the south-eastern area of the continent during the December 2016–April 2017 wet season (Fig. 8), might have prompted a southward spread of Afrotropical wild birds and consequently of the virus[17].

## Discussion

Our study examines in detail the occurrence of multiple virus introductions into the African continent during each of the three intercontinental Gs/GD HPAI H5Nx epidemic waves caused by clades 2.2, 2.3.2.1c and 2.3.4.4-B. We found that the likely origin of the virus varies from one epidemic to another, confirming our previous findings[50]. Virus gene flow from eastern Europe seems to have been the only route of virus spread into the African continent during the first wave, while during the two following epidemics multiple virus origins were identified: the Middle East, East Europe, South and North-Central Asia.

Estimated tMRCAs of viruses from different African regions (Fig. 3) fit well with the timing of reported outbreaks in the African countries (Supplementary Fig. 1). For the earliest wave, the time of the first virus incursion in West Africa dated at least four months before the identification of the first outbreak. Differently, for the second and the third epidemic waves the estimated MRCA ages were close to the first virus discovery, which may suggest an increased capacity of HPAI surveillance and diagnosis.

The different wild bird surveillance efforts among countries and the few or no data from wild birds available from Africa did limit our ability to infer the contribution of domestic and wild birds in the spatial movements to and within Africa at a refined scale. Import of live domestic birds likely from the Middle Eastern and/or South Asian countries might have been the possible cause of virus introduction during the second epidemic wave (2014–2015). Based on the data from the Food and Agriculture Organisations (FAO) (UN-FAO: faostat3.fao. org, accessed on 24 October 2018), West African countries reported import of live chickens and ducks in 2014–2015 from East Asia and Europe only. However, we cannot exclude unreported cross-border trades with the Middle East and South Asia, which was occasionally recorded in other periods (UN-FAO: faostat3.fao.org, accessed on 24 October 2018). By contrast, during the first (2005–2006) and the third (2016–2017) epidemic waves, our analyses indicated that the main route of virus spread into Africa was most probably via infected wild Anseriformes, which include many migratory populations. This finding is consistent with satellite-tracked waterfowl studies and experimental infection data[51,52].

On a global level, our analysis clearly suggested a central role of wild birds in the spatial spread of the earliest Gs/GD HPAI H5Nx epidemic wave, for which the largest amount of sequence data was available (Fig. 6). Whereas, wild and domestic birds appeared to have contributed equally to virus diffusion during the second and the third waves. When we tried to explicitly account for the skewed sampling data from wild birds to certain geographic locations, our analyses indicated wild birds as being the major dispersers of the virus at a global level during all the three epidemic waves. We obtained such results by enforcing host species transitions from wild to domestic birds, as being the most likely mode of transmission during a transcontinental AI virus spread[25,27,29–36]. Detections of HPAI H5 viruses in clinically healthy migratory birds[27,34,52–54] and experimental infection data[52,55–58] indicated that several waterfowl species can spread HPAI H5 during the period of asymptomatic infection, making

migratory birds potential candidates for the intercontinental spread of the virus. The key role of long-distance migrants in the dispersal of HPAI H5 viruses has been suggested by several authors based on phylogenetic analyses, epidemiological investigations and on the timing and direction of the intercontinental spreads, which coincided with fall bird migrations[25,28–36]. Moreover, HPAI H5-infected wild species have been reported in a variety of countries before or simultaneously with poultry outbreaks, and direct or indirect contacts with wild birds have been frequently identified as the most probable cause of virus introduction into poultry[12,13,15,26,27,29,35]. In some African countries, illegal poaching of wild birds, which are kept in rural communities and then sold at markets, is not uncommon and may represent a possible bridge between wild and domestic birds[59]. The role of wild birds in the African continent is also supported by the virological and serological evidences of circulation of the H5 subtype in the wild population[51,60], in particular during the most recent epidemic wave when HPAI H5N8 was widely detected in wild bird species in several countries such as Egypt[18], Cameroon[12], Uganda[13] and South Africa[15,17]. Moreover, in all epidemic waves the first outbreaks in Africa were reported between November and January (Supplementary Fig. 1), during or immediately after the fall bird migrations.

The long branches which separate the African viruses from the progenitor ones, exemplified by the long-distance dispersal observed in the phylogeographic analyses, coupled with the lack of overlap between some of the observed gene flows (i.e. the Middle East–West Africa) and migratory flyways/live bird trades might conceal additional spatial movements between the origin and final destination locations. Of note, some West African, Middle Eastern and South Asian countries with high poultry densities, positioned along important migratory flyways and close or neighbouring to countries affected by HPAI H5 reported few or no outbreaks. Whether this reflects the real situation or not is a matter which cannot be assessed. Increased sampling and sequencing efforts and the identification and monitoring of stopover sites along the migratory flyways—which host large congregations of birds from various species, geographic origins and destinations—can help to improve our understanding of the means and routes of virus diffusion and to clarify uneven virus spread among different Africa regions. However, multiple infrastructural (e.g., accessibility of certain wild bird hotspots, laboratory diagnostic capacity) and financial obstacles in the African continent prevent a proper implementation of a true early warning system.

The potential role of wild birds in virus spread to Africa for two of the three epidemic waves raises a critical question to be answered: why did the Gs/GD HPAI H5Nx virus not reach the African continent during the 2009–2010 and 2014–2015 intercontinental epidemic waves of clades 2.3.2.1c and 2.3.4.4 group A? Virus expansion is a complex phenomenon, which is shaped by intricate interplays between avian hosts' ecology, virus properties and climatic variables, such as temperature, humidity and precipitation. As previously reported by Napp et al.[61] and as shown by the anomaly temperature maps in Fig. 7, between October and December 2016 unusually low temperatures affected north-central Asia and eastern Europe. These two regions were identified as the possible 2.3.4.4-B viral sources for Egypt and are on the migratory flyways connecting the Palaearctic and north Africa[26]. The cold weather might have favoured the southward spread of wild birds, and consequently of the virus to warmer areas. Similarly, the 2005–2006 cold winter in Europe, which Ottaviani et al.[43] suggested as having influenced the virus movement across the continent, might also have enhanced southward virus diffusion. To date,

there is no evidence that cold weather in the temperate regions may affect wild bird migration patterns into Sub-Saharan Africa; however, knowledge of variation of trans-continental wild bird migration patterns in response to changing ecological conditions is scarce.

West Africa has been the most important point of virus introduction for all the three epidemic waves and has played the most important role in the virus spread within the continent. West Africa is rich in large permanent wetlands, such as the Senegal River delta, the Inner Niger delta, the Middle Niger Floodplains and Lake Chad, which are important wintering grounds for several migratory ducks[62], and is located at the crossroads of two migratory flyways—the Black Sea/Mediterranean and the East Atlantic. This is also one of the African regions with the highest poultry density, which may account for the persistence of the disease in this area. The unique characteristics of this geographic area make West Africa a crucial hotspot for virus introduction and dissemination within the continent and a target region for virus surveillance. Focusing on areas at risks of AIV incursion in West Africa, our study demonstrated that Nigeria played a key role in virus introduction and spread to the other countries during the first and second waves. The results are consistent with previous findings[63] and provide a possible explanation in the eco-epidemiological conditions in this country, characterised by one of the highest poultry densities in the region and by the presence of key wintering sites for migratory birds, such as Hadejia-Nguru Wetlands and Lake Chad[62]. Nigeria is also the country with the largest human population in Africa (about 200 million people) as well as a very high population density (215 people per square km)[64]. This combined with a complex socio-economic situation which limits the government's intervention capacity for disease control, as demonstrated by persistent AIV circulation, may pose an increased risk for the emergence of viruses with pandemic potential.

Egypt, too, emerged as a hotspot for the invasion of multiple lineages during the first and last epidemic waves. This country is rich in wetlands along the Nile River and the Mediterranean and Red Sea coasts, and it harbours four important stopover sites[65] for wild birds that migrate along the East Africa-West Asia, the Mediterranean/Black Sea or the more regional Rift Valley-Red Sea flyways[26]. However, no viral exchange between Egypt and other African countries was observed.

The last epidemic wave witnessed for the first time the spread of the Gs/GD HPAI H5Nx virus to east and southern Africa. Migratory birds that overwinter in east and south Africa generally breed in eastern Europe and central Asia and migrate southwest through the Black Sea-Caspian region and the Middle East, while most birds overwintering in West Africa breed in west and central Europe[62]. The fact that both east and South African viruses were related to West African strains might be a consequence of (i) multiple introductions of related viruses from distinct but partially overlapping flyways, (ii) within Africa wild bird migrations, or (iii) poultry trade between African countries. Our data indicate that intra-Africa wild bird migration appears to be the most likely cause of virus spread. Afrotropical waterfowl movements are complex and are mainly driven by the availability of food and water[62], as demonstrated for some trans-equatorial migrant species[39,40]. The abundant rainfall during the 2016–2017 Sub-Saharan and southern African rainy seasons, which created temporary wetlands, attracting a large number of birds, coupled with the anomalous drought that affected central Africa in October–December 2016, might have shaped the intra-African movements of wild birds during this period.

Thanks to a well established network involving thirteen African research institutions, we collected a comprehensive genetic and epidemiological dataset on the continent. This allowed us to reveal the central role of wild migratory birds in virus introduction into Africa for two of the three epidemic waves. We also identified the regions at high risk of virus introduction and spread, such as West Africa, and recommend that these regions should be prioritised for wild and domestic bird surveillance and enhancement of biosecurity.

The Gs/GD emergence and spread has taught us that uncontrolled circulation of avian influenza in any region could become a threat at any latitude and longitude.

Understanding the implications that climate change might have on wild bird migration, and identification of the most vulnerable regions for AIV emergence have become a top priority to improve our ability to fight AIV. This is particularly true for emerging economies in Africa, where co-circulation in domestic birds of multiple Gs/GD HPAI H5 clades and different AIV subtypes (H5N1/H5N8/H9N2)[66–68], combined with poor surveillance, limited response capacity and deficient reporting, creates the opportunity for strains with unexpected zoonotic potential to appear and spread.

## Methods

**Genome sequencing and generation of consensus sequences**. Within the scope of this study, we generated complete genomes for 40 African AIVs. Total RNA was purified from 37 HPAI H5N1 and 3 HPAI H5N8 positive clinical samples using the QIAsymphony DSP Virus/Pathogen Kits, in combination with the QIAsymphony SP (Qiagen). Complete influenza A virus genomes were amplified with the SuperScript III One-Step RT-PCR system with Platinum Taq High Fidelity kit (Invitrogen, Carlsbad, CA) and one pair of primers complementary to the conserved elements of the influenza A virus (MBTUni-12-DEG 5′-GCGTGATCAGCRA AAGCAGG-3′ and MBTUni-13 5′-ACGCGTGATCAGTAGAAACAAGG-3′)[69]. Sequencing libraries were obtained using Nextera XT DNA Sample preparation kit (Illumina, San Diego, CA, USA) following the manufacturer's instructions and quantified using the Qubit dsDNA High Sensitivity kit (Invitrogen, USA). The average fragment length was determined using the Agilent High Sensitivity Bioanalyzer Kit. The indexed libraries were pooled in equimolar concentrations and sequenced in multiplex on an Illumina MiSeq instrument using a 2 × 250 bp paired-end [PE] mode, according to the manufacturer's instructions.

Illumina read quality was assessed using FastQC v0.11.2. Raw data were filtered by removing: (i) reads with more than 10% of undetermined (N) bases; (ii) reads with more than 100 bases with Q score below 7; and (iii) duplicated paired-end reads. Remaining reads were clipped from Illumina Nextera XT adaptors with scythe v0.991 (https://github.com/vsbuffalo/scythe) and trimmed with sickle v1.33 (https://github.com/najoshi/sickle). Reads shorter than 80 bases or unpaired after previous filters were discarded. High quality reads were aligned against a reference genome using BWA v0.7.12[70]. Alignments were processed with Picard-tools v2.1.0 (http://picard.sourceforge.net) and GATK v3.5[71–73] in order to correct potential errors, realign reads around indels in respect of the reference genome and recalibrate base quality. Single nucleotide polymorphisms were called using LoFreq v2.1.2[74] and the outputs were used to generate the consensus sequences. For this study, we focused our analyses on the *HA* gene.

**Datasets design**. For the global analysis, *HA* gene sequence data and relative epidemiological information of avian HPAI H5Nx viruses with a minimum sequence length of 1500 bp, from Africa, Asia and Europe were retrieved from the Global Initiative on Sharing All Influenza Data (GISAID) platform and GenBank for each of the clade considered in this study: 2.2 (2005–2011, 1514 sequences); 2.3.2.1c (2009–2017, 621 sequences); and 2.3.4.4-B (2013-2018, 511 sequences). To assess the robustness of our analysis and mitigate sampling bias, we assembled three different datasets for each clade, each containing about 240–250 sequences (Supplementary Datas 1–9). Three different subsampling strategies were used based on: (1) virus epidemiological information (sampling location, collection date, host)—epi-based dataset, (2) phylogenetic diversity (http://www.cibiv.at/software/pda/)—tree-based dataset, and (3) randomly down-sampling sequences—random dataset. More detail on the subsampling procedure and dataset composition is provided in the Supplementary Methods.

For the local analysis of the African continent, we collected all the available African *HA* sequences for each clade under investigation, except for the Egyptian viruses of clade 2.2. This clade has been circulating in Egypt since the end of 2005 and more than 800 *HA* sequences (>1500 nt) collected from avian species were available in GISAID. Since these viruses form a single, well-defined monophyletic group, we included in our analysis only 11 randomly selected sequences collected during the first

years of the epidemic (2005–2008). Each dataset contains a total of 196 (clade 2.2), 210 (clade 2.3.2.1c) and 77 (clade 2.3.4.4-B) sequences (Supplementary Datas 10–12).

Details on sequencing and composition of each dataset are provided in the Supplemental Methods.

The *HA* sequences of each generated dataset were aligned through the Multiple Alignment using Fast Fourier Transform (MAFFT) programme version 7[75].

**Missing data assessment**. To assess possible bias in the outputs of our analyses, we determined the proportion of sequence data available with respect to the reported outbreaks in the geographic area and period of time considered in this study. To this end, we retrieved data on HPAI H5N1 and H5N8 outbreaks from 2005 to 2018 from Asia, Europe and Africa from the Empres-i animal disease information database kept by the Food and Agriculture Organisation of the United Nation (FAO)[19], and the *HA* sequence data and respective epidemiological information from the Global Initiative on Sharing All Influenza Data (GISAID) platform (accessed on 18 July 2018).

**Bayesian evolutionary inference**. All Markov chain Monte Carlo (MCMC) sampling analyses were performed using BEAST v1.8.4 package[76] in combination with BEAGLE library to improve computational performance[77]. We employed an uncorrelated lognormal relaxed molecular clock that allows for rate variation across lineages. The HKY85 + $\Gamma_4$ model with two partitions (1st + 2nd positions vs. 3rd position), base frequencies and $\Gamma$-rate heterogeneity unlinked across all codon positions (the SRD06 substitution model)[78] was used along with a Bayesian skygrid coalescent tree prior. For viruses for which only the year or month of virus collection was available, the lack of tip date precision was accommodated by sampling uniformly across a 1-year or 1-month window[79]. MCMC chains were run for at least 100–250 million iterations, and mixing and convergence properties of the chains were assessed using Tracer v1.6, with statistical uncertainty reflected in values of the 95% highest posterior density (HPD). MCC trees were summarised using TreeAnnotator v1.8.4 after the removal of an appropriate burn-in, and the trees were visualised using FigTree v1.4.2 (http://tree.bio.ed.ac.uk/software/figtree/).

We estimated spatial diffusion dynamics among a set of geographic regions, ranging from 6 to 12 depending on the dataset, using a Bayesian discrete phylogeographic approach[80]. We used a non-reversible continuous-time Markov chain model and incorporated Bayesian stochastic search variable selection (BSSVS) to focus on a sparse set of rate parameters[80]. Bayes factor (BF) support for individual transitions between discrete locations was computed using Spread D3 v0.9.6[81]. We interpreted the strength of statistical support as follows: positive support for 5 < BF < 20, strong support for 20 < BF < 150 and very strong support for BF > 150.

As a complementary approach to discrete phylogeographic inference, we also estimated the HPAI H5NX diffusion dynamics in continuous space[82]. A strict Brownian diffusion model that assumes a homogeneous rate of diffusion was tested against relaxed random walk models that allow dispersal rates to vary along branches. The best-fitting model was selected using the path sampling and stepping-stone sampling marginal likelihood estimators as implemented in BEAST[83,84]. The reconstructed dispersal history was visualised using Spread D3 v0.9.6[81].

To explore the role of different avian host populations (domestic Galliformes, domestic Anseriformes, wild Anseriformes and other wild bird species) in the expansion dynamics of the three distinct HPAI H5NX clades, we capitalised on the epi-based data sets to incorporate both a continuous spatial diffusion process and a discrete host transmission process in a single Bayesian analysis[31]. Although both processes are modelled independently, the joint inference allows us to summarise host-specific contributions to the spatial dispersal dynamics and to estimate the host-specific diffusion rate. To this end, we mapped the complete host trait history in the posterior tree distribution and condition on this to delineate host-specific trajectories in the phylogeographic history as implemented in BEAST v1.8.4[83,84]. For the delineated host-specific trajectories in the posterior tree distribution, we summarised the realisations of the continuous spatial diffusion process. The number of available samples from poultry is generally higher than from wild birds. We attempted to minimise the impact of this sampling heterogeneity by imposing a unidirectional virus flow from wild to domestic birds. The results obtained when imposing this constraint were compared to those generated when allowing for all possible host species transitions.

**Reporting summary**. Further information on research design is available in the Nature Research Reporting Summary linked to this article.

## Data availability
The authors declare that the main data supporting the findings of this study are available within the article and its Supplementary Information files. The *HA* sequences generated in this study were deposited in the Global Initiative on Sharing All Influenza Data database (GISAID). Accession numbers of the sequences downloaded from the public databases GenBank (https://www.ncbi.nlm.nih.gov/nucleotide/) or GISAID (https://www.gisaid.org) or obtained in this study are listed in the Supplementary Datas 1–12. Extra data are available from the corresponding authors upon request.

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

## Acknowledgements

We wish to thank Francesca Ellero at the IZSVe, Legnaro (PD), Italy, for her assistance in facilitating the exchange of information and sample submissions and Paola Mozzi for her help in making Fig. 3. We extend our sincere appreciation to all the operators engaged in field sampling who made the present investigation possible. We acknowledge the authors and the originating and submitting laboratories of the sequences from the GISAID EpiFlu Database on which this research is partly based (Supplementary Data 1 to 12). Partial support for this work was provided by UN-FAO with funding from USAID under the OSRO/GLO/501/USA and OSRO/GLO/507/USA projects and by European Union's Horizon 2020 research and innovation programme under grant agreement No 727922 (DELTAFLU). We thank OIE for supporting training activities of the NVRI staff on genetic characterization of the Nigerian H5 avian influenza viruses through the OIE Twinning Project (IZSVe/NVRI) entitled "Improving NVRI laboratory capacity for better control of the Avian Influenza virus at National and Regional level". The research leading to these results has also been partly funded by the European Research Council under the European Union's Horizon 2020 research and innovation programme (grant agreement no. 725422-ReservoirDOCS). P.L. acknowledges support by the Research Foundation – Flanders FWO, G066215N, G0D5117N and G0B9317N). B.V. is a post-doctoral research fellow supported by the FWO.

## Author contributions

I.M. designed the project. A.F., I.M., B.Z., B.V. and P.L. wrote the paper. I.M., A.F., B.Z., B.V. and P.L. conceived and designed the analyses. A.M. and G.Z. generated the data used in the analyses. A.F. and B.Z. analysed the data. B.V. and P.L. supervised the analyses. C.A., R.A., A.A.l., A.A.r., J.A.A., E.C.H., M.B.C., N.G., E.G.M., T.J., S.D.J., G.M., C.M., M.S.M., D.B.N., I.S., A.T., A.W., L.W., A.Y.P., G.Z., A.M. critically reviewed the manuscript. C.A., R.A., A.Al., J.A.A., E.C.H., M.B.C., E.G.M., T.J., S.D.J., G.M., C.M., M.S.M., D.B.N., I.S., A.T., A.W., L.W., A.Y.P. provided samples and/or genetic data.

## Competing interests

The authors declare no competing interests.
