## [Peer Review File · Nature Communications]

Reviewers' comments:

Reviewer #2 (Remarks to the Author):

Fusaro et al sequenced the 40 complete genomes of H5 highly pathogenic avian influenza viruses from Africa and analyzed their HA sequences with those of the related viruses from other regions to shed lights into the African virus' origin. State-of-the-Art phylodynamics analyses have been done to understand the virus transmission dynamics in geographical scale and among host species. The authors found that Africa is mostly the dead-end for the virus introduction, and the introductions were mainly facilitated by wild bird migration and poultry trading.

Major comments:

The data analyses and figures were very well done. The only concern I have is the enforcement of host species transitions from only wild to domestic birds in the joint host-geo phylodynamics analysis, which is not biologically reasonable. And in such enforcement, wild birds are 'forced' unavoidably occupied the majority of ancestral host state estimates with bias. It seems when such enforcement was taken out, the result became not significant. I do not feel this biased enforcement is an appropriate way to resolve the bias from the high number of domestic samples. I think random sub-sampling to achieve more balanced number between domestic and wild samples is a better workaround.

It seems that this joint host-geo analysis is a major part providing the evidence for the authors' interpretation that introduction to Africa is contributed by mainly bird trade and wild bird migration. To make such interpretation more solid, have the authors more formally tested the contribution of bird trade and migration flyways/pattern to the virus diffusion in the phylodynamics framework using e.g. GLM?

Minor comments:

A graph/table summarizing all H5 HPAI outbreaks reported in Africa, particularly their timing and duration would be useful (epi data). This also will provide further support to which African countries are the first entry point of the virus introduction. Will be useful to compare these epidemiological timing and the tMRCA timing shown in Figure 3.

It will be useful to readers by providing literature that suggested the spread of avian influenza virus was facilitated by wild bird migrations.

Did the authors performed analysis of the internal gene sequences of those sequenced African H5 HPAIVs? Any interesting/consistent findings?

Reviewer #3 (Remarks to the Author):

The authors seek to describe the role Africa plays in the transmission of highly pathogenic avian influenza. This work represents an impressive international collaboration and is a valuable contribution to our understanding of spatial dynamics of avian flu in Africa, a region that broadly remains understudied in infectious disease epidemiology. Overall, the paper is very well written. After some minor changes, I recommend that this work be accepted for publication.

My primary critique is that the work would benefit from more discussion of the possible consequences of a complicated observation process. Drawing robust conclusions from imperfect data is difficult, and from the methods the authors have used, it is clear that this is on their minds.

The authors remark that in regions with better surveillance of wild bird populations, 42-66% HPAI H5Nx outbreaks are in wild birds, but, for example, in West Africa, only 1% of outbreaks are reported in wild birds. This almost certainly represents dramatic underreporting; what consequence might that have for this analysis? Similarly, are there regions of Africa where no HPAI avian flu data are available, but are known to have large domestic and/or wild bird populations? Please address the potential effects of biases in the observation process in the discussion.

Minor comments:

In main text Figure 6, I suggest using slightly thicker lines in the top three panels, and in the legends, consistently use the bold legend colors found both at the bottom of Fig 6 and in Supplementary Figure 8. If the legend currently at the bottom of Fig 6 were placed in between the two rows, it probably would not need to be repeated four times in the figure. Currently, the pale grey and blue colors are difficult to distinguish on some monitors.

Supplementary Figure 8: x-axes are unlabeled/do not have units.

Reviewer 2

Fusaro et al sequenced the 40 complete genomes of H5 highly pathogenic avian influenza viruses from Africa and analyzed their HA sequences with those of the related viruses from other regions to shed lights into the African virus' origin. State-of-the-Art phylodynamics analyses have been done to understand the virus transmission dynamics in geographical scale and among host species. The authors found that Africa is mostly the dead-end for the virus introduction, and the introductions were mainly facilitated by wild bird migration and poultry trading.

Author's response: We wish to thank the reviewer for the careful review of our paper and for the useful suggestions, which increase the strength of our findings and improve the quality of the paper.

Major comments:

The data analyses and figures were very well done. The only concern I have is the enforcement of host species transitions from only wild to domestic birds in the joint host-geo phylodynamics analysis, which is not biologically reasonable. And in such enforcement, wild birds are 'forced' unavoidably occupied the majority of ancestral host state estimates with bias. It seems when such enforcement was taken out, the result became not significant. I do not feel this biased enforcement is an appropriate way to resolve the bias from the high number of domestic samples. I think random sub-sampling to achieve more balanced number between domestic and wild samples is a better workaround.

Author's response: We agree with the reviewer on the importance to have a balanced number between wild and domestic samples. Those included in the database defined in the text of the manuscript as "Epi-based" were selected in order to have roughly equitable numbers of sequences for each host, geographic category and year of collection (see Supplementary material, lines 57-60). In the table below (Table A), we reported for each genetic clade and each geographic region the percentage of HA sequences of HPAI H5NX collected from wild birds i) in the dataset containing all the sequences available in the public databases (GenBank and GISAID) at the time of writing (TOT) and ii) in the EPI-BASED dataset. This shows that the sampling is already fairly balanced in terms of total sequences for domestic and wild birds for each clade (percentage of wild bird sequences ranges from 42.5% for clade 2.2 to 46.06% for clade 2.3.4.4B), but it is very heterogeneous per region, particularly for Africa that is of course the location of interest in our study. Unfortunately, this is an issue that cannot be addressed using further downsampling, since no or just a few sequences from wild birds are available from this geographic area.

Table A. Percentage of wild bird sequences of clades 2.2, 2.3.2.1c and 2.3.4.4-B, for each geographic region identified in this study, in the complete datasets (TOT), which contain all sequences available in the public databases for each clade, and in the Epi-based dataset (EPI-BASED).

LOCATION	% wild bird sequences					
	2.2		2.3.2.1c		2.3.4.4-B	
	TOT	EPI-BASED	TOT	EPI-BASED	TOT	EPI-BASED
Central Africa	0	0	/	/	25	25
Central Asia	84.29	75	100	100	96.30	87.50

East Asia	40	36.36	33.64	44.94	57.14	33.33
East EU	53.33	57.69	25	50	47.37	39.39
Middle East	2.08	20.83	50	25	52.63	36.36
South Asia	3.11	20	0	0	100	100
West Africa	1.67	5.88	0	0	0	0
West EU	90	77.14	/	/	66.43	52.38
South Africa	/	/	/	/	20	66.67
TOT	0.23	42.50	0.25	42.53	0.41	46.06

We are aware and agree with the reviewer that enforcement of host species transitions from wild to domestic birds may to some extent be conceived as unrealistic since not all transmissions are from wild to domestic, but we use it to prevent what is an even more unrealistic scenario because of the lack of wild bird sampling. Moreover, transmission of HPAI H5 viruses from domestic to wild birds has been rarely reported in comparison to the reverse transmission (Global Consortium for H5N8 and Related Influenza Viruses, 2016; Trovao et al., 2015; Olsen et al., 2006; Hill et al., 2015; Poen et al., 2019; Alarcon et al., 2018; Lee et al., 2017; Si et al., 2009; Verhagen et al., 2015; Ndumu et al., 2018, Abolnik et al., 2018; Mulatti et al., 2018; Naguib, 2019; Wade 2018; Bevins et al., 2016; Jeong et al., 2014).

To further address this issue, we now attribute more importance to the unconstrained analyses, which have been moved from the supplementary material to the main text (Fig. 6a, lines 295 to 303). Moreover, we have now provided a more in depth explanation of the reasons that motivated our choice to perform analyses with this constraint and we have more clearly stated and discussed the limits of such choice (lines 304 to 311 in the “Results” and lines 430-440 in the “Discussion”).

It seems that this joint host-geo analysis is a major part providing the evidence for the authors’ interpretation that introduction to Africa is contributed by mainly bird trade and wild bird migration. To make such interpretation more solid, have the authors more formally tested the contribution of bird trade and migration flyways/pattern to the virus diffusion in the phylodynamics framework using e.g. GLM?

Author’s response: We appreciate the Reviewer’s suggestion to test the contribution of bird trade and migration flyways to the virus diffusion using a GLM parameterization of the discrete diffusion model. Unfortunately, data related to poultry trade among countries are far from complete and the migration routes are very heterogeneous among wild bird species. Despite these limitations, we carried out the suggested analyses, which are here below reported for the Reviewer’s convenience:

Methods

Poultry trade data

Official data on the international trade of live poultry (quantity of chickens, turkeys and ducks) from 2005 to 2016 (the most recent year of data available) were downloaded from the Food and Agriculture Organization of the United Nations (UN-FAO) (<http://www.fao.org/faostat/en/#data/TM>, accessed on July 4, 2019). Years 2005-2008, 2009-2016 and 2016 for clade 2.2, 2.3.2.1c and 2.3.4.4-B, respectively, were summed up to estimate the international trade during the period of diffusion and intercontinental spread of each clade. For each clade, only countries reporting the outbreaks were selected. Because of the inconsistencies in data reporting from different countries, the quantity of both exports and imports was compared for each pair of countries and only the maximum trade quantity was recorded. Countries were aggregated into nine geographical regions - Central Africa, West Africa, South Africa, East

Europe, West Europe, Middle East, North-Central Asia, East Asia, South Asia – consistent with the geographic regions used for the discrete phylogeographic analyses.

Wild bird flyways data

Defining wild bird flyways requires an extreme simplification of the great complexity of bird migration. Migration routes and schedules can vary by species, but also by population within species and between individuals in the same population. Migration can also vary according to the age and/or sex of the bird, to the season and to the weather (Boere and Stroud, 2006). As a consequence, attempts to make any generalization will necessarily imply losing out on relevant information, which can be potentially misleading. Within Africa, wild bird migration can be even more complex, and is driven mainly by food availability in relation to rainfall. Several different migratory behavioral types were observed, such as: short-distance migrants, ‘rain’ migrants, nutrition migrants, post-breeding dispersers, altitudinal migrants and so on (Dodman and Diagana, 2006).

The term flyway has been defined by Boere & Stroud (2006) as “the entire range of a migratory bird species (or groups of related species or distinct populations of a single species) through which it moves on an annual basis from the breeding grounds to non-breeding areas, including intermediate resting and feeding places as well as the area within which the birds migrate”. At the global level generalized flyways have been described for the wader or shorebird group (Boere & Stroud, 2006), for which eight flyways have been recognized: the East Atlantic Flyway, the Mediterranean/Black Sea Flyway, the West Asia/Africa flyway, the Central Asia/Indian sub-continent Flyway, the East Asia/Australasia Flyway, and three flyways in the Americas and the Neotropics (Boere & Stroud, 2006). Although any simplification will necessarily imply missing some information, this classification works well for a majority of waterbirds. Moreover, these flyways have been adopted in a large number of studies to demonstrate the contribution of wild birds on the spread of AI (Lycett et al., 2019; Xu et al., 2016; Trovao et al., 2015; Wang et al., 2008; Tian et al., 2015; Olsen et al., 2006). Thus, we decided to use them for testing the contribution of flyways to virus spread. However, it should be taken into account that in Eurasia there is an important component of east-west migration, such as from Eastern to Western Palearctic or from Nearctic to Western Palearctic, which are not well captured in this flyway model (Wetlands International, 2012).

Phylogeographic inference with epidemiological predictors

To assess the contribution of potential explanatory variables (predictors) of the viral global diffusion process on phylogeographic reconstructions, we used of the generalized linear model (GLM) extension of Bayesian discrete phylogeographic models (Lemey et al., 2014) available in BEAST 1.10.2. This allows the reconstruction of the spatial diffusion history throughout the tree and simultaneously evaluates the contributions of various potential predictors.

Using this approach, we tested the following predictors: i) geographic distance (the distance between the centroid of each geographic region), (ii) live poultry trade (the quantity of poultry exchanged between each pair of geographic regions), (iii) the spread along a migratory flyway. Specifically, for each flyway (East Atlantic Flyway, the Mediterranean/Black Sea Flyway, the West Asia/Africa flyway, the Central Asia Flyway, the East Asia/Australasia Flyway), we specified a predictor that has 1 for rates between geographical regions within that flyway and 0 for those which are not. Each flyway included the following regions:

- East Atlantic Flyway: Central Asia, South Africa, West Africa, West Europe
- Mediterranean/Black Sea Flyway: Central Africa, Central Asia, East Europe, Middle East, West Africa, West Europe
- West Asia/Africa flyway: Central Africa, Central Asia, Middle East, South Africa
- Central Asia Flyway: Central Asia, South Asia

- *East Asia/Australasia Flyway: Central Asia, East Asia*

For each clade, only regions affected by the epidemic wave were included in the flyway. The analyses were performed using the Epi-based dataset of each clade. In addition, the effect of including sample size as a predictor was also examined.

To eliminate the effect of the magnitude of different predictors, all of them (except binary predictors) were log-transformed and standardized.

Bayes factors (BFs) were calculated to determine the support for the inclusion of each predictor in the model.

Results

In the 2.2 and 2.3.4.4-B datasets, only geographic distance yielded strong BF support (Table B), indicating that distance was an important driver of location transitions. It consistently made a negative contribution to the virus spread, meaning that migration is inferred to be stronger between countries that are geographically closer to each other. This predictor was not supported ($BF < 3$) for clade 2.3.2.1c.

None of the other tested predictors - poultry trade and inclusion in a migratory flyway - resulted in any noticeable support ($BF < 3$) by any of the analysed datasets, with and without considering the number of samples by location as a distinct predictor.

The fact that higher poultry trade turned out not to be associated with poultry spread may be suggestive that human activities are possibly not the major driver of HPAI H5 virus spread, although we cannot exclude this may be a consequence of incomplete data available on movement of live animals between countries or of illegal poultry trade.

As mentioned above, testing flyways as a predictors is challenging, given that i) migratory behaviors ranging from sedentary birds to nomadic and short/long distance migrants is known to be highly variable (Gaidet et al., 2010); ii) different migratory routes are used by different species; iii) several intersections between flyways exist, making virus transfer between flyways possible, iv) migratory behavior can change according to environmental conditions (i.e. cold weather, drying of wetlands, temporarily available habitat), short-term weather patterns and longer-term climate change (Boere & Stroud, 2006); v) long-distance migratory flights are interspersed with relatively long periods of staging at stopover-sites located along a migratory flyway, where birds of various species, geographic origin and destinations aggregate in large numbers (Gaidet et al., 2010). In addition, the gap of sequence data from some of the key sites along the major flyways (i.e. the area around the Caspian Sea) may have affected our results. Thus, the lack of support of flyways as potential predictors is likely a consequence of the challenges associated with including such information in the model, since the contribution of wild birds in the HPAI H5 virus dispersal have been documented in several studies (Gaidet et al., 2010; Global Consortium for H5N8 and Related Influenza Viruses, 2016; Olsen et al., 2006; Hill et al., 2015; Poen et al., 2019; Alarcon et al., 2018; Lee et al., 2017; Trovao et al., 2015).

In order to avoid bringing a misleading message associated with the absence of support for bird flyways and poultry trade in our analyses, which would distract the attention from our main findings, we have opted not to include these analyses in our manuscript.

Table B. Bayes factors of tested predictors

Predictors	Bayes Factors		
	2.2	2.3.2.1c	2.3.4.4-B

Poultry trade	2.710	0.035	0.068
Geographic distance	76.857	0.085	76.857
Mediterranean/Black Sea Flyway	0.136	0.087	0.184
Central Asia Flyway	0.593	0.176	0.208
East Asia Flyway	0.366	0.103	1.261
East Asia/Australasia Flyway	0.350	0.144	0.362
West Asia/Africa flyway	0.946	0.129	0.210

Minor comments:

A graph/table summarizing all H5 HPAI outbreaks reported in Africa, particularly their timing and duration would be useful (epi data). This also will provide further support to which African countries are the first entry point of the virus introduction. Will be useful to compare these epidemiological timing and the tMRCA timing shown in Figure 3.

Author's response: According to the reviewer's suggestion, a graph summarizing all the African outbreaks for each clade has been included in the supplementary information (Supplementary Figure 1). The discussion now also includes a comparison between the estimated tMRCA of each virus introduction and the time of the reported outbreaks (lines 411 to 416).

It will be useful to readers by providing literature that suggested the spread of avian influenza virus was facilitated by wild bird migrations.

Author's response: According to the reviewer suggestion, we included in the text more references suggesting the contribution of wild birds in the HPAI H5 spread and expanded our discussion (lines 441 to 454)

Did the authors performed analysis of the internal gene sequences of those sequenced African H5 HPAIVs? Any interesting/consistent findings?

Author's response: We analysed the internal genes for all the African viruses sequenced in our laboratory and identified some reassortant viruses.

Intra-clade reassortment events among viruses belonging to distinct co-circulating sub-lineages of the same genetic clade have been reported in Nigeria, where we identified five distinct gene constellations during the 2006-2007 epidemic of clade 2.2 (Cattoli et al., 2009; Fusaro et al., 2010) and nine in 2015-2016 during the 2.3.2.1c epidemic (Laleye et al., 2018). Similarly, in Cameroon, all the identified H5N1 viruses of clade 2.3.2.1c were reassortants between two sublineages (WA-1 and WA-2) co-circulating in West Africa (Wade et al., 2018).

In addition to these intra-clade reassortments, the progenitor of the 2.3.2.1c African viruses was generated from intersubtype reassortment between H5N1 (segments 2 to 8) and H9N2 (segment 1) viruses of Asian origin. These viruses have the same gene constellation of a H5N1 human isolate identified in Canada from a patient who had returned from China (Monne et al., 2015; Tassoni et al., 2016), suggesting that the reassortment likely occurred in China.

The two genetic groups of African H5N8 viruses of clade 2.3.4.4-B identified in the HA phylogeny showed different gene constellations. Specifically, one group, including viruses from Cameroon,

Niger, Nigeria, Republic of Congo and Uganda clusters with viruses collected in Asia, Europe and Egypt, while the second group, including viruses from Cameroon, Niger and South Africa possess the highest similarity with a H5N8 virus from India, except for the NP which grouped with a H5N8 virus from the Russian Federation (Wade et al., 2018; Ndumu et al., 2018; Twabela et al., 2018).

Since the characteristics of the complete genomes of the African viruses have been already described in previous papers, we have decided to not repeat these descriptions in this study.

REFERENCES

- Abolnik, C. et al. The incursion and spread of HPAI H5N8 Clade 2.3.4.4 within South Africa. <https://doi.org/10.1637/11869-042518-Reg.1> (2018). doi:10.1637/11869-042518-REG.1*
- Alarcon, P. et al. Comparison of 2016–17 and Previous Epizootics of Highly Pathogenic Avian Influenza H5 Guangdong Lineage in Europe. *Emerg. Infect. Dis.* 24, 2270–2283 (2018).*
- Bevins, S. N. et al. Widespread detection of highly pathogenic H5 influenza viruses in wild birds from the Pacific Flyway of the United States. *Sci. Rep.* 6, 1–9 (2016).*
- Boere, G.C. & Stroud, D.A. The flyway concept: what it is and what it isn't. Waterbirds around the world. Eds. G.C. Boere, C.A. Galbraith and D.A. Stroud. The Stationery Office, Edinburgh, UK. pp. 40-47 (2006)*
- Cattoli, G.. et al. Highly pathogenic avian influenza virus subtype H5N1 in Africa: a comprehensive phylogenetic analysis and molecular characterization of isolates. *PLoS One.* 4(3):e4842 (2009)*
- Dodman, T. & Diagana, C. . Conservation dilemmas for intra-African migratory waterbirds. in *Waterbird around the world* (ed. G.C. Boere, C. A. G. & D. A. S.) 218–223 (The Stationery Office, 2006).*
- Fusaro, A. et al. Evolutionary dynamics of multiple sublineages of H5N1 influenza viruses in Nigeria from 2006 to 2008. *J Virol.* 84(7):3239-47 (2010)*
- Gaidet, N. et al. Potential spread of highly pathogenic avian influenza H5N1 by wildfowl: Dispersal ranges and rates determined from large-scale satellite telemetry. *J. Appl. Ecol.* 47, 1147–1157 (2010).*
- Global Consortium for H5N8 and Related Influenza Viruses. Role for migratory wild birds in the global spread of avian influenza H5N8. *Science* (80-.). 354, 213–217 (2016).*
- Hill, S. C. et al. Wild waterfowl migration and domestic duck density shape the epidemiology of highly pathogenic H5N8 influenza in the Republic of Korea. *Infect. Genet. Evol.* 34, 267–277 (2015).*
- Jeong, J. et al. Highly pathogenic avian influenza virus (H5N8) in domestic poultry and its relationship with migratory birds in South Korea during 2014. *Vet Microbiol.*173(3-4):249-57 (2014)*
- Laleye, A. A two-year monitoring period of the genetic properties of clade 2.3.2.1c H5N1 viruses in Nigeria reveals the emergence and co-circulation of distinct genotypes. *Infect Genet Evol.* 2018 Jan;57:98-105. doi: 10.1016/j.meegid.2017.10.027.*
- Lee, D.H. et al. Evolution, global spread, and pathogenicity of highly pathogenic avian influenza H5Nx clade 2.3.4.4. *J Vet Sci.* 31;18(S1):269-280 (2017)*
- Lemey, P. et al. Unifying viral genetics and human transportation data to predict the global transmission dynamics of human influenza H3N2. *PLoS Pathog* 10:e1003932 (2014).*

- Lycett, S.J. *et al.* A brief history of bird flu. *Philos Trans R Soc Lond B Biol Sci.* 24;374(1775):20180257 (2019)
- Monne, I *et al.* Highly Pathogenic Avian Influenza A(H5N1) Virus in Poultry, Nigeria, 2015. *Emerg Infect Dis.* 21(7):1275-7 (2015)
- Mulatti, P. *et al.* Integration of genetic and epidemiological data to infer H5N8 HPAI virus transmission dynamics during the 2016-2017 epidemic in Italy. *Sci Rep.* 21;8(1):18037 (2018)
- Naguib, M. M. *et al.* Avian influenza viruses at the wild–domestic bird interface in Egypt. *Infect. Ecol. Epidemiol.* 9, 1575687 (2019).
- Ndumu, D. *et al.* Highly pathogenic avian influenza H5N8 Clade 2.3.4.4B virus in Uganda, 2017. *Infect. Genet. Evol.* 66, 269–271 (2018).
- Olsen, B. *et al.* Global patterns of influenza A virus in wild birds. *Science* 312, 384–8 (2006)..
- Poen, M. J. *et al.* Co-circulation of genetically distinct highly pathogenic avian influenza A clade 2.3.4.4 (H5N6) viruses in wild waterfowl and poultry in Europe and East Asia, 2017-18. *Virus Evol.* 5, vez004 (2019).
- Si, Y. *et al.* Spatio-temporal dynamics of global H5N1 outbreaks match bird migration patterns. *Geospat. Health* 4, 65–78 (2009).
- Tassoni, L. *et al.* Genetically Different Highly Pathogenic Avian Influenza A(H5N1) Viruses in West Africa, 2015. *Emerg. Infect. Dis.* 22, 2132–2136 (2016).
- Tian, H. *et al.* Avian influenza H5N1 viral and bird migration networks in Asia. *Proc Natl Acad Sci U S A.* 2015 Jan 6;112(1):172-7. doi: 10.1073/pnas.1405216112
- Trovão, N. S., Suchard, M. A., Baele, G., Gilbert, M. & Lemey, P. Bayesian Inference Reveals Host-Specific Contributions to the Epidemic Expansion of Influenza A H5N1. *Mol. Biol. Evol.* 32, msv185 (2015).
- Twabela, A. T. *et al.* Highly Pathogenic Avian Influenza A(H5N8) Virus, Democratic Republic of the Congo, 2017. *Emerg. Infect. Dis.* 24, 1371–1374 (2018).
- Verhagen, J. H., Herfst, S. & Fouchier, R. A. M. Infectious disease. How a virus travels the world. *Science* 347, 616–7 (2015).
- Wade, A. *et al.* Highly Pathogenic Avian Influenza A(H5N8) Virus, Cameroon, 2017. *Emerg. Infect. Dis.* 24, 1367–1370 (2018).
- Wang, G. *et al.* H5N1 avian influenza re-emergence of Lake Qinghai: phylogenetic and antigenic analyses of the newly isolated viruses and roles of migratory birds in virus circulation. *J Gen Virol.* 89(Pt 3):697-702 (2008)
- Wetlands International, 2012. *Waterbird Population Estimates, Fifth Edition. Summary Report.* Wetlands International, Wageningen, The Netherlands
- Xu, Y. *et al.* Southward autumn migration of waterfowl facilitates cross-continental transmission of the highly pathogenic avian influenza H5N1 virus. *Sci Rep.* 10;6:30262. (2016)

Reviewer 3

The authors seek to describe the role Africa plays in the transmission of highly pathogenic avian influenza. This work represents an impressive international collaboration and is a valuable

contribution to our understanding of spatial dynamics of avian flu in Africa, a region that broadly remains understudied in infectious disease epidemiology. Overall, the paper is very well written. After some minor changes, I recommend that this work be accepted for publication.

Author's response: We wish to thank the reviewer for his/her positive feedback and useful suggestions.

My primary critique is that the work would benefit from more discussion of the possible consequences of a complicated observation process. Drawing robust conclusions from imperfect data is difficult, and from the methods the authors have used, it is clear that this is on their minds. The authors remark that in regions with better surveillance of wild bird populations, 42-66% HPAI H5Nx outbreaks are in wild birds, but, for example, in West Africa, only 1% of outbreaks are reported in wild birds. This almost certainly represents dramatic underreporting; what consequence might that have for this analysis? Similarly, are there regions of Africa where no HPAI avian flu data are available, but are known to have large domestic and/or wild bird populations? Please address the potential effects of biases in the observation process in the discussion.

Author's response: Following the reviewer's suggestion, we modified the discussion in order to emphasize the challenges of this study, mainly associated to the differences in the implementation of wild bird surveillance between countries and the few or no data from wild birds available from Africa. We have also provided a more detailed explanation of the motivation for our choice to enforce host species transitions from wild to domestic birds, and related to this, we have moved the unconstrained host analyses from the supplementary material to the main text (Fig. 6 has been modified; lines 295 to 311; 417 to 419; 430 to 440).

We have also included more references throughout the text, thus suggesting the contribution of wild birds to the HPAI H5 spread, which may help interpreting our results. The Discussion has also been expanded accordingly (lines 441 to 454).

Some West African, Middle Eastern and South Asian countries with high poultry densities, positioned along important migratory flyways and close or neighbouring countries affected by HPAI H5, reported few or no HPAI H5 outbreaks. Whether this reflects the real situation or not is a matter that cannot be assessed. However, the long branches which separate the African viruses from the progenitor ones, exemplified by some long-distance dispersal in the phylogeographic analyses, coupled with the lack of overlap between some of the observed gene flows (i.e. the Middle East-West Africa) and migratory flyways or live bird trades clearly indicate a gap in the data. This data has now been included in the discussion (lines 455 to 468).

Minor comments:

In main text Figure 6, I suggest using slightly thicker lines in the top three panels, and in the legends, consistently use the bold legend colors found both at the bottom of Fig 6 and in Supplementary Figure 8. If the legend currently at the bottom of Fig 6 were placed in between the two rows, it probably would not need to be repeated four times in the figure. Currently, the pale grey and blue colors are difficult to distinguish on some monitors.

Author's response: Supplementary Figure 8 has been removed and all the graphs have been moved within Figure 6, which has been modified according to the reviewer's suggestions.

Supplementary Figure 8: x-axes are unlabeled/do not have units.

Author's response: Thank you for spotting this oversight. We have now labeled all the axes in all the graphs moved in Figure 6.

REVIEWERS' COMMENTS:

Reviewer #2 (Remarks to the Author):

The authors have sufficiently addressed my previous questions. And I do not have further comments.

Reviewer #3 (Remarks to the Author):

While I am concerned about how strong biases in reporting might affect the conclusions drawn in this work, I nonetheless believe this is a valuable contribution and am generally satisfied with the revisions the authors have made.

REVIEWERS' COMMENTS:

Reviewer 2:

The authors have sufficiently addressed my previous questions. And I do not have further comments.

Author's response: *We wish to thank the reviewer for the time spent for revising the manuscript and the positive feedback.*

Reviewer 3:

While I am concerned about how strong biases in reporting might affect the conclusions drawn in this work, I nonetheless believe this is a valuable contribution and am generally satisfied with the revisions the authors have made.

Author's response: *We wish to thank the reviewer for the time spent for revising the manuscript and the positive feedback.*